# Magnolol as a Protective Antioxidant Alleviates Rotenone-Induced Oxidative Stress and Liver Damage through MAPK/mTOR/Nrf2 in Broilers

**DOI:** 10.3390/metabo13010084

**Published:** 2023-01-04

**Authors:** Weishi Peng, Nanxuan Zhou, Zehe Song, Haihan Zhang, Xi He

**Affiliations:** 1College of Animal Science and Technology, Hunan Agricultural University, Changsha 410128, China; 2Hunan Engineering Research Center of Poultry Production Safety, Changsha 410128, China

**Keywords:** magnolol, rotenone, liver damage, broilers, MAPK/mTOR/Nrf2 pathway

## Abstract

This study aimed to investigate the protective effects and molecular mechanism of magnolol supplementation on rotenone-induced oxidative stress in broilers. Two hundred and eighty-eight old male AA broilers were randomly divided into four groups: the CON group: basic diet with sunflower oil injection; the ROT group: basic diet with 24 mg/kg BW rotenone; the MAG + ROT group: basic diet with 300 mg/kg magnolol and rotenone injection; and the MAG group: basic diet with 300 mg/kg magnolol and sunflower oil injection. At 21–27 days of age, the broilers in each group were intraperitoneally injected with rotenone (24 mg/kg BW) or the same volume of sunflower oil. The results showed that magnolol reversed the decrease in ADG post-injection and FBW via rotenone induction. Compared to the ROT group, MAG + ROT group enhanced the average daily gain post injection (*p* < 0.05). Magnolol supplement could improve the activity and mRNA expression of rotenone-suppressed antioxidant enzymes such as GSH and GSH-PX (*p* < 0.05). Similarly, the MDA content as an oxidative damage marker was significantly reduced after magnolol addition (*p* < 0.05). The hepatocyte apoptosis and the mRNA expression of apoptosis-related signaling pathway in the ROT group increased, but magnolol supplementation inhibited rotenone-induced apoptosis through the Nrf2 signaling pathway. Through RNA transcriptome analysis, there were 277 differential genes expressions (DEGs) among the CON group with ROT group, and 748 DEGs were found between the ROT group and the MAG + ROT group. KEGG pathway enrichment found that magnolol relieved rotenone-induced energy metabolism disorder and oxidative damage through signaling pathways such as MAPK and mTOR. In conclusion, magnolol attenuates rotenone-induced hepatic injury and oxidative stress of broilers, presumably by restoring hepatic antioxidant function via the MAPK/mTOR/Nrf2 signaling pathway.

## 1. Introduction

In the past few decades, intensive selection procedures throughout genetic programs have rapidly increased the daily weight gain, shortened the breeding time, and, at the same time, increased the risk of broilers exposed to oxidative stress [1]. Under practical management procedure of broiler breeding, many stressors, such as heat/cold stress, toxins, or pathogen infection, have been addressed in relation to elevating oxidative stress [2]. Oxidative stress is the accumulation of free radicals due to the imbalance between oxidation and antioxidation [1]. It can reduce the production performance of poultry and shorten the shelf life of poultry products, causing huge economic losses for the poultry industry [3]. Therefore, reducing oxidative stress during production may play a beneficial role in improving growth performance or meat quality of broiler.

In recent years, plant extracts have been used as a potential method to protect poultry from oxidative stress [4,5]. Magnolol (MAG), a plant extract, is a major phenolic substance extracted from the roots and bark of Magnolia officinalis [6], which has extensive antioxidant effects due to its special phenolic hydroxyl structure [7,8]. Previous studies also found that magnolol can effectively eliminate free radicals in animals, enhance the activity of antioxidant enzymes, and improve the growth performance and antioxidant capacity of yellow feather broilers [7]. Therefore, diet supplementation with MAG may be an important measure to alleviate the oxidative stress of broilers in production. However, there are few studies on MAG in broilers at present, and the specific mechanism of its antioxidant effect has not been reported.

As an insecticide widely used in agriculture, Rotenone (ROT) is a respiratory chain inhibitor that inhibits mitochondrial respiratory chain complex, which is used to induce oxidative stress in animal models [9]. It will lead to electron leakage and release a large amount of reactive oxygen species into the mitochondrial matrix to destroy the structure, leading to oxidative stress [10]. Therefore, this study established an oxidative stress model via intraperitoneal injection of ROT to explore the improvement of MAG supplementation in the diet and its related mechanisms, so as to provide a theoretical basis for MAG to alleviate oxidative stress in broiler production.

## 2. Materials and Methods

### 2.1. Animals and Experimental Design

The protocols used in the animal experiments were approved by the Institutional Animal Care and Use Committee at Hunan Agricultural University (File code: HAU ACC 2022024). A total of 288 one-day-old male Arbor Acres (AA) broiler chicks were randomly divided into 4 groups, which were designed as a 2 × 2 factorial arrangement that included a diet factor (the chicks were fed either a basal diet or a diet supplemented with 300 mg/kg magnolol [95% purity, Milcota Ltd., Changsha, China] of diet from 1 to 29 day of age) and a stress factor (injected with either rotenone or solution) at a dose of 24 mg/kg BW or an equivalent amount of sunflower oil at 21 to 27 day of age [11,12]. Each of these groups consisted of 6 replicate cages of 12 broilers each. Broilers in the oxidative stress model group were intraperitoneally injected with rotenone for 7 consecutive days. Samples were collected 48 h after the end of the injection period.

During the experiment, the chicks were kept in pens with dimensions of 140, 80, and 40 cm in length, width, and height. All chicks were kept under the same management guidelines and the environment was kept at a temperature of 34 °C for the first week, gradually dropping to 24 °C by the fourth week and afterwards. The light schedule was 23 h of light and 1 h of dark. Feed and fresh water were freely used during the trial periods. The ventilation of the house and birds’ vaccination status followed the commercial recommendation of this breed. According to the recommended standards formulated by the National Research Commission (1998) for broilers, the nutrient constitution of basic diet was prepared as shown in Table 1.

### 2.2. Sample Collection

At 29 day of age, two birds from each cage were randomly selected for sampling after a 12 h feed withdrawal period. Blood samples were collected by slicing the jugular vein. Serum was then separated after centrifugation for 10 min at 3500 rpm at 4 °C, and immediately frozen at −20 °C pending analysis. Liver was divided into 2 portions. One was divided for histopathological examination; the other slice was immediately frozen in liquid nitrogen and stored at −80 °C for biochemical and mRNA assays.

### 2.3. Growth Performance Measurement

All broilers were weighed individually after a 12 h feed withdrawal period on 1 and 21 day, and cumulative intake of feed was recorded per replicate to determine average daily gain (ADG), average daily feed intake (ADFI), and feed to gain ratio (F/G) during 1 to 21 day of age. During the injection period, the broilers were weighed daily, and the feed consumption in this period was accounted for. The average daily gain and average feed intake during injection were calculated.

### 2.4. Oxidative Parameters Determination

Total antioxidant capacity (T-AOC), catalase (CAT), glutathione (GSH), glutathione peroxidase (GSH-PX), total superoxide dismutase (T-SOD), malondialdehyde (MDA), and aspartate amino-transferase (AST) assay kits were procured from Nanjing Jiancheng Bioengineering Institute (Nanjing, China). The levels of MDA and activity of T-AOC, CAT, GSH, GSH-Px, T-SOD, and AST in serum and liver were detected. Furthermore, the total protein content of organization was determined by the method of coomassie brilliant blue using the commercial kit (Nanjing Jiancheng Bioengineering Institute, Nanjing, China) in accordance with the manufacturer’s instructions. 8-hydroxy-2-deoxyguanosine (8-OHdG) was detected using an ELISA kit (mlbio, ml059825, China).

### 2.5. Histopathological Examination

Samples from the left lobe of the liver were obtained, fixed in 4% paraformaldehyde 48 h, and embedded in paraffin blocks. Paraffin sections were dehydrated using ethanol (Sinopharm Chemical Reagent Co., Ltd.,100092683, Shanghai, China) and stained with hematoxylin–eosin and observed (servicebio, G1005, Wuhan, China) under a light microscope (NIKON DS-U3) for assessment of histopathological damage (AiFang biological, Changsha, China).

### 2.6. Apoptotic Status Analysis

The liver samples were divided into 5 mm sections for subsequent apoptotic analysis, which was carried out with One Step TUNEL Apoptosis Assay Kit (Beyotime, C1088, China) staining. Tissue sections were permeabilized with proteinase K (Beyotime, ST538, China) at 37 °C for 22 min. After washing with phosphate-buffered solution three times, the TUNEL mixed reagents were added to the sections and incubated in dark conditions at 37 °C for 2 h. We used DAPI (Beyotime, C1002, China) to label the nuclei. The TUNEL-positive cells were visualized using a fluorescent microscope (Nikon Eclipse C1, Nikon, Tokyo, Japan) and were defined by a green color with the DAPI label.

### 2.7. qReal-Time PCR Analysis

Total RNA and their reverse transcription of liver were isolated using the SteadyPure Universal RNA Extraction Kit (Accurate Bioengineering Co., Ltd., Changsha, China). The mRNA expression levels of CAT, SOD, GPX, Nrf2, Keap1, HO-1, NQO1, Bcl-2, Bax, caspase-3, and XIAP were measured using a quantitative real-time PCR (RT-qPCR) technique with the primers. The RT-qPCR were performed using the SYBR Green Premix Taq (Accurate Bioengineering Co., Ltd., Changsha, China). The mRNA levels were calculated using the 2^−ΔΔCT^ method. The primer sense and antisense sequences were listed in Table 2.

### 2.8. Transcriptome Sequencing

The total RNA from broiler jejunal tissue was extracted by Trizol. The RNA was purified, fragmented, reverse transcribed, and amplified. The library fragments were enriched using PCR and quality control was performed. The fragments were then sequenced on Illumina platform. The broiler reference gene (Gallus gallus. Ensembl_release106. GRCg6a.) was searched using Genome in NCBI (its Path was ft http://asia.ensembl.org/Gallus_gallus (accessed on 3 November 2022)), and the genome database was referenced to Ensembl. Filtered Reads were compared to the Gallus gallus reference genome using HISAT2 software. Differential statistical analysis of gene expression was performed by DESeq. Venn maps of differentially expressed genes were drawn using the R language ggplots2 package to show gene distribution, gene expression ploidy differences, and significance results. Two-way clustering analysis of the concatenated sets and samples of differential genes was performed using the R language Pheatmap package, and the Euclidean method was used to calculate the distance. Complete Linkage was used for clustering. GO enrichment analyses were performed with the database established by the Gene Ontology Consortium (http://geneontology.org/ (accessed on 3 November 2022)). KEGG enrichment analyses were performed with the database of Kyoto Encyclopedia of Genes and Genomes (http://www.kegg.jp/ (accessed on 3 November 2022)). Sequencing services were provided by Personal Biotechnology Co., Ltd. Shanghai, China. The data were analyzed using the free online platform Personalbio Genes Cloud (https://www.genescloud.cn (accessed on 3 November 2022)).

### 2.9. Statistics Analysis

All the above data were expressed by means ± SD, and each group of data was analyzed by two-way analysis of variance (ANOVA) via the general linear model procedure of IBM SPSS Statistics 25.0 software. Statistical significance was considered as *p* < 0.05.

## 3. Results

### 3.1. Growth Performance

In Table 3, the growth performance of broilers on 1–21 day had no significance before injection (*p* > 0.05). However, the FBW and ADG post injection of rotenone-induced broilers were dramatically lower than CON group after ROT injection (*p* < 0.05). Meanwhile, the ADG post injection of the MAG + ROT group was no difference with control group, but presented a significant difference compared with the ROT group (*p* < 0.05).

### 3.2. Antioxidant Index

The effect of dietary MAG supplementation and ROT injection on the oxidative parameters of serum and liver are shown in Table 4. Compared to the CON group, rotenone challenge decreased plasma and liver CAT activity and increased AST accumulation (*p* < 0.05). Injection of rotenone dramatically increased the content of MDA, whereas it decreased GSH and GSH-PX concentrations in the serum (*p* < 0.05). In contrast, broilers that consumed a magnolol-supplemented diet showed a significant reduction in the MDA content (*p* < 0.05), and robustly improved the GSH and GSH-PX contents after rotenone injection (*p* < 0.05). Moreover, feeding a diet with magnolol to broilers obviously increased the concentrations of GSH in the serum compared with those fed a basal diet (*p* < 0.05). In liver, the broilers challenged with rotenone showed a sharp increase in the content of 8-OHdG (*p* < 0.05), while the activities of T-AOC and T-SOD significantly decreased (*p* < 0.05). In addition, magnolol treatment alone did not improve the concentration of GSH-PX in the liver, but significantly increased GSH-PX content in broiler exposure to rotenone (*p* < 0.05).

### 3.3. Histopathological Observation

The liver histopathological sections of the four groups were shown in Figure 1A. The broiler accompanied by rotenone injection led to the infiltration of inflammatory cell, hepatic cord disorder and hepatocytes sizes variation compared with the CON group. Nevertheless, magnolol supplementation exerted an opposite effect on the vacuoles and inflammatory cell infiltration compared to the ROT group. In Figure 1B, hepatocyte apoptosis was examined. Compared to the CON group, the apoptosis degree of liver cells significantly increased after ROT injection. On the contrary, the liver damage caused by ROT injection significantly recovered after MAG was added to the diet. In addition, the broilers fed a diet with magnolol showed a lower content of hepatic apoptosis and better liver cell morphology (Figure 1A,B).

### 3.4. Relative Expression of Gene

The mRNA abundance of antioxidant enzymes, Nrf2 signaling pathway and apoptosis-related genes was shown in Table 5. Compared with the unchallenged broilers, rotenone induction lowered the mRNA abundance gene expressions of GPX (*p* = 0.043). However, dietary magnolol treatment reversed the gene expression of GPX after rotenone challenge (*p* < 0.05). Likewise, a magnolol diet without rotenone injection made no difference on the expression levels of Nrf2, Keap1, HO-1, and NQO1 compared to the CON group, but in the MAG + ROT group, broilers challenged with rotenone had a higher mRNA expression of Nrf2 and Keap1. Moreover, magnolol supplementation exerted a marked opposite effect, decreasing the mRNA abundance of Bcl2 and increasing the mRNA abundance of Bax via rotenone injection (*p* < 0.05). In addition, the Bcl2 expression was upregulated by magnolol treatment (*p* < 0.05).

### 3.5. RNA Sequencing

Transcriptomic sequencing of chicken liver was examined to study differentially expressed genes associated with oxidative stress. The principal components analysis (PCA) was shown in Figure 2A, and the expression levels of the four groups were distributed in different locations. In Figure 2B, compared to the CON group, broilers challenged with rotenone had 135 upregulated and 142 downregulated differentially expressed genes (DEGs). Similarly, a total of 748 DEGs between the ROT group and MAG + ROT group were shown in Figure 2C, which was composed of 667 downregulated and 81 upregulated genes. Then, KEGG was used to understand the oxidative-stress-related functions of these differentially expressed genes induced by rotenone. Firstly, KEGG enriched features of the amino acid metabolism such as alanine, aspartate, glutamate, cysteine, methionine, tyrosine and glycine, and degradation of valine, leucine, and isoleucine between the CON group and the ROT group. Under the same conditions, the pyruvate metabolism and TCA cycle were also enriched (Figure 2D). Compared to the ROT group, broilers with a magnolol supplement after rotenone induction had different expression on the calcium, MAPK, VEGF, and mTOR signaling pathway (Figure 2E).

## 4. Discussion

Magnolol is one of the foremost effective ingredients of the traditional Chinese medicine Magnolia Officinalis, which has received widespread attention due to its anti-inflammatory and antioxidant effects [13]. Previous research has shown that magnolol could increase the growth performance of broilers, ducks, and laying hens via their antioxidant activity [7,14,15]. Therefore, this experiment established an oxidative stress model through intraperitoneal injection of rotenone to explore whether it can improve growth performance. Rotenone is a cytotoxic substance that can directly penetrate the cell membrane. It can inhibit the electron transfer of the mitochondrial respiratory chain, leading to the loss of mitochondrial function, the enhancement of cellular oxidative stress, and ultimately inducing apoptosis [9]. The results showed that intraperitoneal injection of rotenone can significantly reduce the average daily gain and final body weight of broilers. This was consistent with the results of Akinmoladun et al.’s experiments, in which rats were treated with rotenone to significantly reduce weight gain [16]. However, dietary supplementation with MAG could reverse the average daily gain and loss of final body weight after rotenone challenge, which is consistent with the weight loss of stress-relieved mice [17,18].

MAG can alleviate the weight loss of mice under oxidative stress, but what are the reasons and specific mechanisms of this process? To answer these questions, we measured the serum and liver antioxidant enzyme levels. The results showed that rotenone injection increased the content of MDA in serum and 8-OHdG in liver of broilers, but decreased the levels of CAT, GSH, GSH-PX, and T-SOD. Supplementation of MAG in the diet reversed the above results induced by rotenone, indicating that magnolol could reduce oxidative stress induced by rotenone by enhancing the antioxidant enzyme system in vivo. A previous study reported that MAG can protect the antioxidant defect mutants of Saccharomyces cerevisiae from oxidative stress [19]. Similar research showed MAG protects PC12 cells from hydrogen peroxide or 6-hydroxydopamine induced by enhancing catalase [20]. The liver is very vulnerable to different kinds of stress, including oxidative stress [21]. Many risk factors, including toxins, environmental pollutants, and pathogenic bacteria, can disturb hepatic redox homeostasis and elicit oxidative-stress-mediated damage, which in turn results in severe liver injury [22]. Once hepatic antioxidant responses are unable to cope with these challenges, the homeostatic systems gradually deteriorate and lead to a decline in disease resistance [23]. Previous studies have confirmed that antioxidant enzymes play a key role in acute alcoholic liver injury in mice to maintain redox balance [24]. Samarghandian et al. [25] reported that curcumin can improve antioxidant enzymes decreased by immobilization-induced oxidative stress in rat liver. Similar research indicated magnolol can ameliorate oxidative stress and inflammation via steatosis induced by autophagy in HepG2 cells [26]. The results of this experiment showed that MAG could increase the decrease in GPX, HO-1, and Bcl-2 and the increase in Bax mRNA induced by ROT. AST is an important transaminase that is mainly synthesized in the liver. The damage of liver cells will lead to the release of AST into the blood. In this study, rotenone treatment can increase AST levels in serum and liver, which reflects that rotenone treatment may damage liver function. Supplementation of MAG in diet can reduce AST content, indicating that MAG can reduce liver damage caused by oxidative stress.

Thus, we studied the phenotype and mechanism of liver injury. H&E and TUNEL staining showed MAG greatly improved the apoptosis of hepatocytes induced by ROT. These results are consistent with the research results obtained previously [27]. In order to study the molecular mechanism of MAG relieving oxidative stress, transcriptome analysis was performed on broilers after the oxidative stress induced by ROT. The RNA transcriptome was mainly enriched in pathways such as MAPK and mTOR. MAPK is an important transmitter of signals from the cell surface to the nucleus [28], and mTOR is an important regulator of cell growth and proliferation [29]. Similarly, research has demonstrated that flavonoids can reduce inflammation via inhibition of MAPKs and Nrf2 activation [30,31]. Zhao et al. [32] reported that mTOR signaling pathway alleviates I/R damage by regulating oxidative stress. In this research, magnolol supplementation effectively upregulated Nrf2 and NQO1 mRNA abundances in the liver of rotenone-induced broilers, indicating that dietary magnolol may improve liver antioxidant capacity through direct activation of Nrf2 pathway. Similarly, it has previously been reported that some prebiotic components of magnolol used in this research could increase Nrf2 and downstream signal gene expressions in the tissues of animals [15]. This suggested that magnolol may relieve rotenone-induced oxidative stress and liver damage through the MAPK/mTOR/Nrf2 signaling pathway. In addition, the KEGG enrichment results of CON group and ROT group showed that rotenone-induced oxidative stress would lead to the disturbance of energy metabolism, resulting in oxidative stress.

These results indicate that rotenone can inhibit the respiratory chain and lead to oxidative stress caused by energy metabolism disorder of broilers, resulting in liver damage and decreased growth performance, and these negative effects can be alleviated by magnolol supplement.

## 5. Conclusions

In summation, our study indicated that dietary magnolol supplementation of 300 mg/kg of magnolol can alleviate the growth retardation and liver damage of broilers. The potential mechanisms for its antioxidant effect may be related to regulation of the gene expression of antioxidant enzyme and factors via the MAPK/mTOR/Nrf2 signal pathway. These findings may provide strategies to researchers and industry stakeholders in preventing oxidative stress for broilers. Further research is warranted to evaluate the usage and elucidate the underlying mechanisms.

## Figures and Tables

**Figure 1 metabolites-13-00084-f001:**
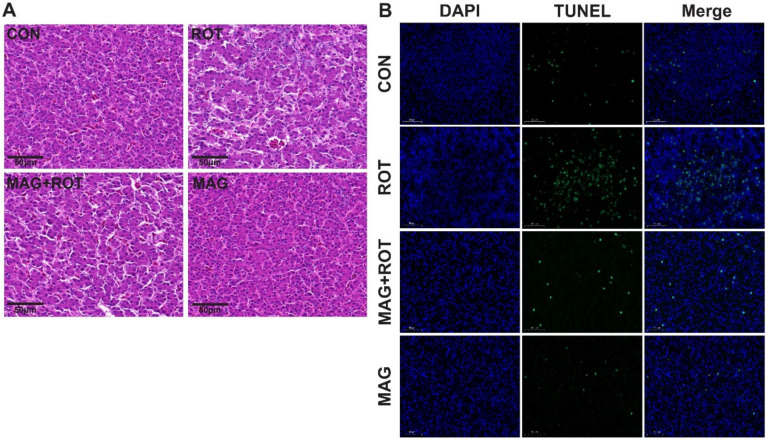
Effect of dietary magnolol supplementation on histopathological examination and apoptosis of liver in broilers challenged with rotenone. (**A**) Representative micrographs of H&E staining of the liver tissue. All images were viewed at the magnification of 400× through a microscope. (**B**) Representative micrographs of TUNEL staining of the liver tissue. All images were viewed at the magnification of 630× through a microscope. Abbreviations: CON—birds were injected sunflower oil with a basal diet; ROT—birds were injected rotenone with a basal diet; MAG—birds were injected rotenone with a 300 mg/kg magnolol diet.

**Figure 2 metabolites-13-00084-f002:**
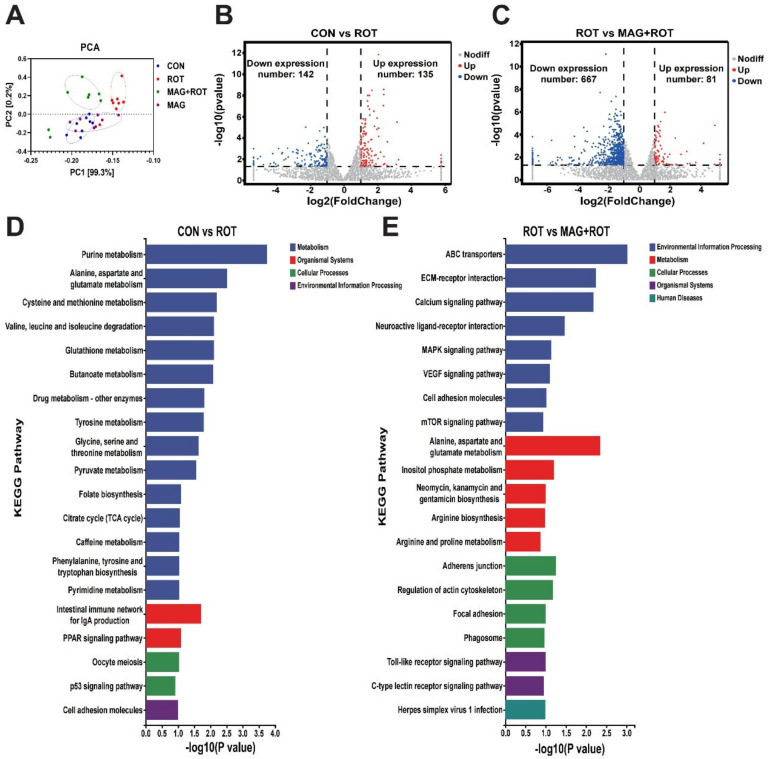
Effect of magnolol on transcriptome of broilers induced by rotenone. (**A**) Principal component analysis for all groups. Volcano plot of differentially expressed genes in jejunum tissues of the CON group and ROT group (**B**), ROT group and MAG + ROT group (**C**). The x—axis and y—axis indicate log2 (fold change) and 2log10 (FDR) of differentially expressed genes in jejunum tissues, respectively. The red color represents upregulated genes, and the blue color represents downregulated genes. Top 20 enriched KEGG terms of differentially expressed genes in the CON group were compared with ROT group (**D**) and ROT group compared with MAG + ROT group (**E**).

**Table 1 metabolites-13-00084-t001:** Composition and nutrient level of the basal diet.

Ingredients	Content, %
Corn	56.51
Soybean meal	35.30
Soybean oil	4.50
Lysine (55%)	0.35
Methionine	0.16
Threonine	0.06
Nacl	0.30
Calcium hydrogen phosphate	1.00
Stone powder	1.52
Premix ^1^	0.30
Total	100.00
Calculated nutrient levels	
Metabolizable energy (Kcal/kg)	3039.94
Crude protein, %	20.65
Calcium, %	0.90
Total phosphorus, %	0.54
Lysine, %	1.27
Methionine, %	0.47
Threonine, %	0.83

^1^ The premix provided the following per kg of diets: Fe, 80 mg, Mn, 120 mg, Se, 0.3 mg, Zn, 110 mg, I, 1.25 mg, VA, 12,000 IU, VD_3_, 5000 IU, VE, 80 IU, VK_3_, 3.2 mg, VB_1_, 3.2 mg, VB_2_, 8.6 mg, VB_6_, 4.3 mg, VB_12_, 0.017 mg, nicotinic acid, 65 mg, pantothenic acid, 20 mg, biotin, 0.22 mg, folic acid, 2.2 mg.

**Table 2 metabolites-13-00084-t002:** The primers and sequences of RT-PCR.

Gene ^1^	Gene Bank ID	Primer Sequence, Sense/Antisense	Length
SOD	NM_205064.2	TTGTCTGATGGAGATCATGGCTTC TGCTTGCCTTCAGGATTAAAGTGA	98
CAT	NM_001031215.1	GTTGGCGGTAGGAGTCTGGTCT GTGGTCAAGGCATCTGGCTTCTG	182
GPX	NM_001163245.2	CAAAGTTGCGGTCAGTGGA AGAGTCCCAGGCCTTTACTACTTTC	136
Nrf2	NM_001396905.1	GGGACGGTGACACAGGAACAAC GCTCTCCACAGCGGGAAATCAG	97
Keap1	XM_015274015	GCATCACAGCAGCGTGGAGAG GGCGTACAGCAGTCGGTTCAG	118
NQO1	NM_001277620.2	CCCGAGTGCTTTGTCTACGAGATG ATCAGGTCAGCCGCTTCAATCTTC	107
HO-1	NM_205344.2	GCTGGGAAGGAGAGTGAGAGGAC GCGACTGTGGTGGCGATGAAG	107
Bcl-2	NM_205339.3	GCTGCTTTACTCTTGGGGGT CTTCAGCACTATCTCGCGGT	128
Bax	XM_422067	GGTGACAGGGATCGTCACAG TAGGCCAGGAACAGGGTGAAG	108
Caspase-3	NM_204725.2	TGGTGGAGGTGGAGGAGC TGTCTGTCATCATGGCTCTTG	183
XIAP	NM_204588.3	ACCAAAAGAAAGCCCACT CATTCCTTACAAGCAACC	147
β-actin	NM_205518.2	CTCTGACTGACCGCGTTACT TACCAACCATCACACCCTGAT	172

^1^ Abbreviations: SOD—superoxide dismutase; CAT—catalase; GPX—glutathione peroxidase; Nrf2—nuclear factor erythroid 2–related factor 2; Keap1—kelch-like ECH-associated protein 1; NQO1—NAD(P)H quinone dehydrogenase 1; HO-1—heme oxygenase 1; Bcl-2—B-cell lymphoma/leukemia 2; Bax—B-cell lymphoma/leukemia 2-associated X protein; XIAP—X-linked inhibitor of apoptosis protein.

**Table 3 metabolites-13-00084-t003:** Effect of dietary magnolol supplementation on the growth performance of broiler under rotenone challenge.

Items	CON		ROT		*p*-Value			
	CON	MAG	CON	MAG	ANOVA ^1^	Diet	Stress	D × S
IBW (g)	41.83 ± 0.35	41.71 ± 0.18	41.65 ± 0.30	41.97 ± 0.42	0.371	0.467	0.757	0.118
21 day BW (g)	864.58 ± 27.48	847.92 ± 24.11	870.14 ± 26.01	865.98 ± 28.92	0.505	0.351	0.291	0.573
1–21 day ADFI (g)	46.18 ± 1.33	46.49 ± 2.58	47.21 ± 2.02	46.86 ± 2.70	0.864	0.980	0.448	0.718
1–21 day ADG (g)	39.18 ± 1.30	38.39 ± 1.15	39.45 ± 1.23	39.24 ± 1.37	0.503	0.344	0.291	0.582
1–21 day F/G	1.18 ± 0.03	1.21 ± 0.09	1.20 ± 0.03	1.19 ± 0.06	0.765	0.553	0.972	0.385
FBW (g)	1463.70 ± 49.50 ^a^	1464.96 ± 36.48 ^a^	1342.85 ± 94.09 ^b^	1414.94 ± 79.10 ^ab^	0.019	0.206	0.006	0.221
ADFI post injection (g)	111.08 ± 7.23	113.50 ± 3.23	101.16 ± 14.03	121.88 ± 36.18	0.367	0.168	0.925	0.271
ADG during injection (g)	74.89 ± 4.11 ^ab^	77.13 ± 4.43 ^a^	59.09 ± 8.85 ^c^	68.62 ± 6.98 ^b^	<0.001	0.035	<0.001	0.178

^1^ Single-factor ANOVA for all treatment groups. ^a–c^ Means in a row without a common superscript letter significantly differ (*p* < 0.05). Abbreviations: IBW—Initial body weight; 21 d BW—21 days body weight; 1–21 day ADFI—average daily feed intake about 1–21 days; 1–21 day ADG—average daily gain about 1–21 days; 1–21 day F/G—feed to gain ratio about 1–21 days; FBW—Final body weight; ADFI during injection—average daily feed intake during injection; ADG during injection—average daily gain during injection; CON—birds were supplemented with a basal diet; ROT—birds were injected with 24 mg/kg BW rotenone at 21 to 27 day of age; MAG—birds were supplemented with a diet with 300 mg/kg magnolol; D × S—the interaction of stress and diet effects.

**Table 4 metabolites-13-00084-t004:** Effect of dietary magnolol supplementation on the oxidative parameters in serum and liver under rotenone challenge.

Items	CON		ROT		*p*-Value			
	CON	MAG	CON	MAG	ANOVA ^1^	Diet	Stress	D × S
Serum								
T-AOC (U/mL)	0.10 ± 0.02 ^ab^	0.11 ± 0.02 ^a^	0.09 ± 0.02 ^b^	0.09 ± 0.01 ^ab^	0.040	0.149	0.023	0.633
CAT (U/mL)	106.54 ± 2.97 ^a^	104.18 ± 2.87 ^a^	100.61 ± 3.90 ^b^	103.56 ± 5.51 ^ab^	0.007	0.793	0.006	0.025
GSH (μmol/L)	4.03 ± 0.84 ^b^	4.62 ± 0.42 ^a^	3.03 ± 0.49 ^c^	3.89 ± 0.66 ^b^	<0.001	0.001	<0.001	0.508
GSH-PX (U/mL)	2178.84 ± 165.91 ^ab^	2265.22 ± 149.00 ^a^	1803.58 ± 210.47 ^c^	2020.65 ± 234.57 ^b^	<0.001	0.255	<0.001	0.010
T-SOD (U/mL)	224.41 ± 38.63	254.39 ± 43.91	209.03 ± 43.78	211.63 ± 35.15	0.123	0.254	0.047	0.336
MDA (nmol/mL)	4.67 ± 0.94 ^b^	5.22 ± 0.82 ^ab^	5.67 ± 1.14 ^a^	4.49 ± 0.89 ^b^	0.025	0.276	0.638	0.005
8-OHdG (ng/mL)	6.91 ± 0.51	6.87 ± 0.27	7.34 ± 0.37	7.22 ± 0.40	0.061	0.534	0.009	0.777
AST(U/L)	49.52 ± 5.16 ^b^	51.00 ± 7.97 ^b^	61.15 ± 10.88 ^a^	53.35 ± 3.25 ^ab^	0.030	0.263	0.018	0.106
Liver								
T-AOC (U/mgprot)	0.023 ± 0.00 ^a^	0.022 ± 0.00 ^a^	0.016 ± 0.00 ^b^	0.018 ± 0.00 ^b^	<0.001	0.928	<0.001	0.031
CAT (U/mgprot)	95.50 ± 15.20 ^a^	80.39 ± 23.02 ^a^	50.04 ± 8.03 ^b^	51.63 ± 12.02 ^b^	<0.001	0.224	<0.001	0.135
GSH (μmol/gprot)	6.19 ± 0.76	6.32 ± 0.98	5.60 ± 0.95	5.99 ± 0.69	0.470	0.432	0.167	0.685
GSH-PX (U/mgprot)	61.87 ± 5.21 ^a^	54.46 ± 5.99 ^b^	42.15 ± 2.35 ^d^	46.93 ± 6.38 ^c^	<0.001	0.406	<0.001	<0.001
T-SOD (U/mgprot)	39.23 ± 3.89 ^a^	38.04 ± 4.94 ^a^	31.99 ± 2.84 ^b^	33.00 ± 3.48 ^b^	<0.001	0.939	<0.001	0.338
MDA (nmol/mgprot)	0.61 ± 0.10	0.57 ± 0.10	0.67 ± 0.10	0.65 ± 0.06	0.141	0.368	0.034	0.812
8-OHdG (ng/mL)	0.78 ± 0.11 ^b^	0.76 ± 0.07 ^b^	0.89 ± 0.12 ^a^	0.81 ± 0.09 ^a^	0.026	0.108	0.018	0.341
AST(U/gprot)	46.27 ± 9.30 ^c^	50.58 ± 10.95 ^bc^	67.06 ± 14.71 ^a^	61.82 ± 14.44 ^ab^	0.004	0.912	0.001	0.266

^1^ Single-factor ANOVA for all treatment groups. ^a–c^ Means in a row without a common superscript letter significantly differ (*p* < 0.05). Abbreviations: T-AOC—total antioxidant capacity; CAT—catalase; GSH—glutathione; GSH-PX—glutathione peroxidase; T-SOD—total superoxide dismutase; MDA—malondialdehyde; 8-OHdG—8-hydroxy-2-deoxyguanosine; AST—aspartate amino-transferase; CON—birds were supplemented with a basal diet; ROT—birds were injected with 24 mg/kg BW rotenone at 21 to 27 d of age; MAG—birds were supplemented with a diet with 300 mg/kg magnolol; D × S—the interaction of stress and diet effects.

**Table 5 metabolites-13-00084-t005:** Effect of dietary magnolol supplementation on liver gene expression levels in broilers challenged with rotenone.

Items	CON		ROT		*p*-Value			
	CON	MAG	CON	MAG	ANOVA ^1^	Diet	Stress	D × S
CAT	1.00 ± 0.40	0.89 ± 0.33	0.92 ± 0.48	0.55 ± 0.28	0.101	0.084	0.140	0.351
GPX	1.00 ± 0.49 ^c^	1.56 ± 0.52 ^b^	0.69 ± 0.32 ^bc^	2.77 ± 0.61 ^a^	<0.001	<0.001	0.043	0.002
SOD	1.00 ± 0.58 ^b^	1.82 ± 0.74 ^ab^	2.13 ± 1.15 ^a^	2.77 ± 0.92 ^a^	0.020	0.052	0.008	0.803
Nrf2	1.00 ± 0.48 ^b^	0.85 ± 0.37 ^b^	1.61 ± 0.32 ^a^	1.70 ± 0.69 ^a^	0.005	0.864	0.001	0.508
Keap1	1.00 ± 0.51 ^b^	1.04 ± 0.12 ^b^	1.82 ± 0.68 ^a^	1.88 ± 0.44 ^a^	0.007	0.829	0.001	0.963
HO-1	1.00 ± 0.51 ^b^	1.26 ± 0.40 ^ab^	1.79 ± 0.66 ^a^	1.18 ± 0.38 ^b^	0.073	0.371	0.077	0.037
NQO1	1.00 ± 0.52	0.91 ± 0.32	0.68 ± 0.29	1.00 ± 0.61	0.554	0.537	0.528	0.278
Bcl-2	1.00 ± 0.25 ^bc^	1.44 ± 0.30 ^a^	0.80 ± 0.22 ^c^	1.34 ± 0.53 ^ab^	0.009	0.001	0.227	0.706
Bax	1.00 ± 0.25 ^bc^	1.39 ± 0.22 ^a^	1.49 ± 0.15 ^a^	0.98 ± 0.28 ^b^	0.010	0.466	0.619	<0.001
Caspase-3	1.00 ± 0.25	0.91 ± 0.16	1.43 ± 0.31	1.14 ± 0.23	0.217	0.049	0.001	0.282
XIAP	1.00 ± 0.25	0.87 ± 0.20	2.57 ± 0.52	1.70 ± 0.45	0.178	0.003	<0.001	0.022

CON—birds were injected sunflower oil with a basal diet; ROT—birds were injected rotenone with a basal diet; MAG + ROT—birds were injected rotenone with a 300 mg/kg magnolol diet; MAG—birds were injected sunflower oil with a 300 mg/kg magnolol diet. ^1^ Single-factor ANOVA for all treatment groups. ^a–c^ Means in a row without a common superscript letter significantly differ (*p* < 0.05). Abbreviations: D × S—the interaction of stress and diet effects.

## Data Availability

The data presented in this study are available in the main article.

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
