# Peer review of "Magnolol as a Protective Antioxidant Alleviates Rotenone-Induced Oxidative Stress and Liver Damage through MAPK/mTOR/Nrf2 in Broilers"

_metabolites, 2023, doi:10.3390/metabo13010084_

Round 1

Reviewer 1 Report

The paper is dealing with the protective antioxidant potential of magnolol on serum and liver through MAPK/mTOR/Nrf2 in broilers. My major concerns with this paper are:

-Oxidative status of the chicken itself has been almost the exclusive focus of research in poultry production. Thoroughly, in the present experiment, the authors assessed the antioxidant potential of Magnolol (a major phenolic substance extracted from the roots and bark of Magnolia officinalis, as a natural antioxidant  in the broilers. The beneficial effect of different plant bioactives derived from essential oils and extracts of medicinal plants to alleviate the oxidative stress in regular chicken diets are well documented in earlier studies. Most of these work demonstrated that MDA, the main oxidative stress indicator, was reduced and prominent antioxidant enzymes such as SOD, CAT and GSH were improved in experimental diets, animal products and tissues as well as a response to supplementation of selected phytochemicals.  This is the case for Magnolol in the present study under an experimentally generated oxidative stress model using rotenone (ROT). Hence, considering the many stress generator factors under intensive management procedure of commercial broiler setting, output from this study has potential to gather interest of scientific committee and the broiler industry stakeholders.

-First and foremost, for a sound and satisfying hypothesis, it is expected the authors to more clearly state what would the general outcomes from this study be, and how this study will reveal a novelty in this area and how will fulfil the gap in the field. What kind of stressor can evoke elevated oxidative stress under practical management procedure of broiler chicken breeding?  

-Noticeably, in contrast to stated antioxidant potential of magnolol, GSH-PX level in liver was reduced while AST concentration was increased when hens treated with magnolol under the conditions of unchallenged protocol. This questionable that magnolol itself induces oxidative stress under normal management procedure though proved marked benefits as antioxidant when birds were subjected to experimental oxidative stress with rotenone. This requires plausible explanations and justifications.

-Another point to consider is that the authors lack rigor regarding presentation of the housing conditions in the house (size and type of the experimental units-floor pen or cage, number of drinkers and size of the feeders allocated to birds, heating, lighting and ventilating of house, vaccination if applied, etc).  Generally speaking, in terms of technical-analytical procedures regarding preparing the extract of Magnolia officinalis bark and determining its pure magnolol concentration, the paper has certain lacks in scientific rigor. All these should be completed benefiting from the very good examples in the scientific literature.

Suggestions shown above could be beneficial to improve the quality of the work:

-          Page 1, line 32 : “intensive selection procedures throughout the genetic programs” instead of “  the broiler breeding mode of commercialization and intensification”

-          Page, 13, line 54 : replace “alleviate” with “improve”

       -    Page, 14, line 79 : replace “It  suggested” with “This suggest”

Briefly, the concept and experimental approach are sound, and the manuscript written clearly. The experimental work has been done with sufficient rigour to justify the conclusions. The paper has some novelty with analytical measurements on oxidative parameters determination, histopathological examination and gene expression. However, as outlined below, there few items that need to be revised to improve it further. 

Author Response

Dear reviewer,

Many thanks for the insightful comments and suggestions about our paper. We have checked the manuscript and revised it according to the comments. Words in red are the changes we have made in the manuscript. For your comments and suggestions, we have put our responses in the attachment, please check! If you have any question about this paper, please don’t hesitate to let me know.

Thanks again for your reply and wish you everything goes will in your job!

Weishi Peng

Student

Reviewer 2 Report

The manuscript “Magnolol as a protective antioxidant alleviates rotenone-induced oxidative stress and liver damage Through  MAPK/mTOR/Nrf2 in broilers” is interesting however, it requires major revisions before acceptance

Please check comments below   

 1.      Please check title it should be “through” not Through

2.      Line: 2: 288 one day old male

3.      Line 34: Please cite reference

4.      Line 37: Please cite reference

5.      Line 53-56: This need to be rewritten. Why study was conducted? It would be better to replace it with Line 47 -49.

6.      Line 67: Model was, better rewrite this line

7.      Line 69-70? What was the reason?

8.      Line 70: Feed and fresh water was freely used during the trial periods

9.      Line 72: the nutrient constitution of basic diet was prepared as shown in Table 1

10.  Table: What is the unit of Content in Table 1

11.  Please write Chemical manufacturer names and Equipment model numbers in methodology section

12.  Line 95 : This sentence is unclear: The levels of T-AOC, CAT, GSH, GSH-Px, T-SOD, 95 MDA, AST in serum and liver

13.  Line 116: please cite reference

14.  Line 123: are

15.  Line 153: Please discuss the result of growth performance before mentioning significant difference

16.  Please add continuous line number in revised manuscript. Page 7, line 1, change title

17.  Page 7, line 2: are shown

18.  Page 10:Line 2: rephrase

19.  Conclusion title is mission, concluding paragraph need to be rewritten

20.  Manuscript need to be through rechecked for grammatical mistakes 

Author Response

(The authors gave the same response as above.)

Round 2

Reviewer 2 Report

The authors have made changes in the manuscript as per the suggestions and comments.